# EARA: Improving Biomedical Semantic Textual Similarity with Entity-Aligned Attention and Retrieval Augmentation

Ying Xiong[1], Xin Yang[1], Linjing Liu[2], Ka-Chun Wong[2]
Qingcai Chen[13], Yang Xiang[3], Buzhou Tang[13*]

[1] Harbin Institute of Technology, Shenzhen    [2] City University of Hong Kong

[3] Peng Cheng Laboratory

## Abstract

Measuring Semantic Textual Similarity (STS) is a fundamental task in biomedical text processing, which aims at quantifying the similarity between two input biomedical sentences. Unfortunately, the STS datasets in the biomedical domain are relatively smaller but more complex in semantics than common domain, often leading to overfitting issues and insufficient text representation even based on Pre-trained Language Models (PLMs) due to too many biomedical entities. In this paper, we propose EARA, an entity-aligned, attention-based and retrieval-augmented PLMs. Our proposed EARA first aligns the same type of fine-grained entity information in each sentence pair with an entity alignment matrix. Then, EARA regularizes the attention mechanism with an entity alignment matrix with an auxiliary loss. Finally, we add a retrieval module that retrieves similar instances to expand the scope of entity pairs and improve the model's generalization. The comprehensive experiments reflect that EARA can achieve state-of-the-art performance on both in-domain and out-of-domain datasets. Source code is available [1].

## 1 Introduction

Biomedical Semantic Textual Similarity (STS) involves measuring the similarity between biomedical texts, such as scientific articles, clinical notes, and Electronic Health Records (EHRs). This measurement can be utilized to identify the relevant information, reduce excessive redundant information in EHRs (Zhang et al., 2011; O'Donnell et al., 2009; Cohen et al., 2013), and improve the accuracy as well as the efficiency of Evidence-Based Medicine (EBM) practices (Hassanzadeh et al., 2019).

As more biomedical STS datasets with detailed annotations have been published, such

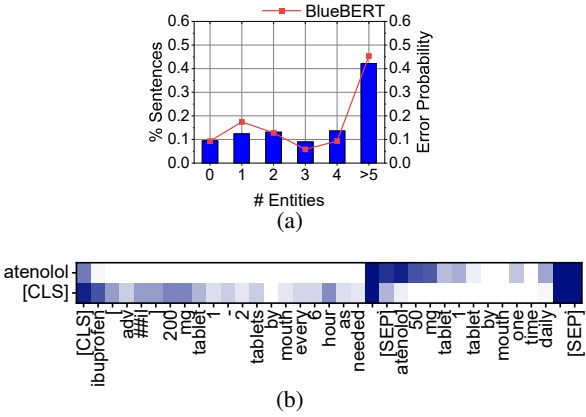

Figure 1: (a) prediction errors of baseline model on N2C2STS datasets and the corresponding entity numbers in sentences. Blue bars represent the proportion of sentences containing the number of entities k (k=0,1,2,3,4,>5) in the sentence; (b) Visualization of the baseline model's attention to the '[CLS]' token and 'atenolol' token of a piece of data in N2C2STS dataset.

as EBMSASS (Hassanzadeh et al., 2019) and N2C2STS (Wang et al., 2020b), a growing number of scholars have paid attention to those STS tasks (Xiong et al., 2020a,b; Chen et al., 2020) and achieved great performance by applying the Pre-trained Language Models (PLMs) (Devlin et al., 2019; Peng et al., 2019; Lee et al., 2020) to those STS tasks.

Unfortunately, naively applying PLMs to biomedical datasets may not yield promising results, owing to the greater semantic complexity of such datasets than those in the general domain. A sentence is deemed semantically intricate when it contains fine-grained information of multiple types, such as entities, which may distract the PLMs. For instance, in Figure 1 (a), we observe a decline in the representational capacity of BlueBERT (as evidenced by an increase in the prediction error ratio) as the number of entities in sentences increases on the N2C2STS datasets. Upon visualizing the at-

---

* Corresponding author.

[1] https://github.com/xy-always/EARA

tention matrix of the '[CLS]' token and 'atenolol' token of a specific instance in the N2C2STS with baseline BlueBERT, we discover that the model tends to focus on punctuation and the sentence containing the selected token.

Besides, during the training process, we also observed that the Pearson Correlation Coefficient (PCC) of models on the training set could often approach 1.0. However, as the PCC of models on the validation set increased, the PCC of models on the test set decreased. The reason might lie in that the size of the biomedical dataset is quite small compared to the size of the general-domain dataset. And the complexity of PLMs is also too excessive for the biomedical dataset, which can easily result in overfitting.

In this paper, we propose EARA, which improves the biomedical semantic textual similarity with *E*ntity-aligned *A*ttention and *R*etrieval *A*ugmentation. EARA first makes PLMs pay more attention to important entities by aligning the same-type entities in sentence pairs. Secondly, EARA builds a knowledge retriever on the MIMIC-III database (Johnson et al., 2016). Thirdly, EARA retrieves similar sentences to expand the scope of entity pairs and feeds them into the entity-aware transformer. Finally, EARA fuses the logits of retrievals and inputs to improve the model's generalization. The experiments show that EARA can achieve state-of-the-art performance on multiple biomedical tasks.

In summary, our work has the following contributions:

- We make the model pay more attention to the same type of entity information by minimizing the L2 loss between the entity alignment matrix and the multi-head attention matrix.

- We add a retriever module and feed retrievals into transformers, which improves the model's generalization and eases the overfitting problem for the biomedical STS datasets.

- We make a comprehensive study on multiple biomedical STS datasets. Results show that our model can achieve promising performance both on in-domain and out-of-domain datasets.

## 2   Related Work

In this section, we mainly introduce the related work from three perspectives: 1) Biomedical STS, 2) PLM-enhanced Models Beyond Biomedical STS and 3) Retrieval-based Methods.

**Biomedical Semantic Textual Similarity** A growing number of works (Chen et al., 2021; Yang et al., 2020) based on PLMs (Devlin et al., 2019; Peng et al., 2019; Lee et al., 2020) have been exploited for biomedical STS datasets, such as N2C2STS dataset (Wang et al., 2020b). For example, Xiong et al. evaluated the pure Siamese CNN (Yin et al., 2016), Siamese RNN (Mueller and Thyagarajan, 2016), and BERT (Devlin et al., 2019) on these biomedical STS datasets, and improved the performance using a gate to fuse the one-hot features representation and deep semantic representation.

There are two lines of recent work to enhance deep learning models on the biomedical STS task. The first used data augmentation strategies (Wang et al., 2020c; Li et al., 2021a) or multi-task learning (Mulyar et al., 2021; Mahajan et al., 2020) to enhance the model's representation. The second introduced external knowledge into the neural network models, which can capture implicit information (Xiong et al., 2020a; Chang et al., 2021). These methods only integrate traditional features and lack interpretations.

**PLM-enhanced Models Beyond Biomedical STS** The PLMs are popularised on other fundamental tasks in NLP and augmented by various knowledge (Bugliarello and Okazaki, 2020; Zhou et al., 2020; Zhang et al., 2020a). Jia et al. (Jia et al., 2020) proposed the Char-Entity-Transformer, which injected lexical information into the char-level BERT and augmented the self-attention using a concatenation of char- and entity-level information. Liu et al. (Liu et al., 2021) proposed a more efficient model than Char-Entity-Transformer by adding a lexicon adapter layer to inject the lexical information into the transformer layer of BERT. Their model achieved the SOTA performance on the named entity recognition, word segmentation, and POS tasks. Stacey et al. (Stacey et al., 2022) supervised the self-attention of BERT using human explanation to the natural language inference tasks. Besides, some works (Zhang et al., 2020b; Li et al., 2021b) used syntax information to train the self-attention module in PLMs and showed promising results on various tasks, like text classification tasks and named entity recognition tasks. Our work leveraged the entity information to regularize the multi-head attention module instead of directly introducing the information.

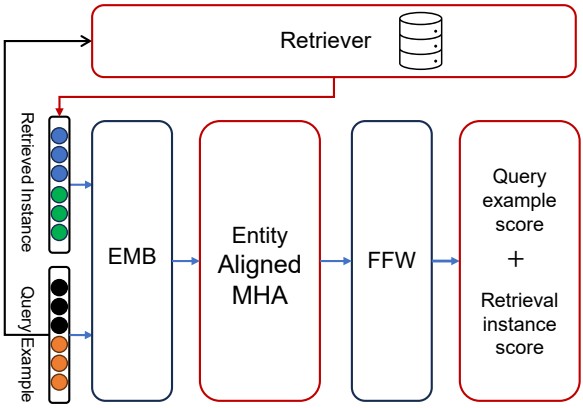

Figure 2: The architecture of our EARA. The dark red rectangles are the newly added or modified modules.

**Retrieval-based Methods** Retrieval-based methods are now being utilized in various NLP tasks. As part of retrieval augmentation research, one approach involved retrieving similar examples from internal training sets to act as demonstrations for prompt learning in a few-shot setting (Gao et al., 2021; Kumar and Talukdar, 2021). Other works used retrieval augmentation for supervised tasks; they aimed to retrieve "answers" for their tasks, such as retrieving the next token in the language model pre-training task, retrieving summarizations from the nearest passages (Chen et al., 2022; Guu et al., 2020; Borgeaud et al., 2022b; Khandelwal et al., 2021; Wang et al., 2022a). While other models rely on nearest neighbours solely for enhancing the prediction process, our approach retrieves similar instances to make the model to consider a broader range of instances, which can help avoid overfitting and improve the models' generalization.

## 3 EARA

### 3.1 Problem Statement

Given two biomedical sentences $S^a = \{x_1^a, x_2^a, x_3^a, ..., x_n^a\}$ and sentence $S^b = \{x_1^b, x_2^b, x_3^b, ..., x_m^b\}$, where $x_i$ is the token in the sentences, $n$ and $m$ stands for the sentence's length of $S^a$ and $S^b$, respectively. The goal of the biomedical STS task is to train a regressor with biomedical STS datasets and to predict the similarity score of two sentences. Specifically, the gold similarity score ranges from 0 to 5 in N2C2STS, where a higher score means the given two sentences are more semantically similar.

### 3.2 Multi-Type Fine-Grained Entity Information

We use cTAKES(Savova et al., 2010) to extract entity types and totally collect 12 types of default entity information. We extract 9 types of entity information, including 'SignSymptomMention', 'Predicate', 'DiseaseDisorderMention', 'MedicationMention', 'RomanNumeralAnnotation', 'AnatomicalSiteMention', 'FractionAnnotation', 'ProcedureMention', and 'DATE'. We add 'O' (standing for outside) and regard the special token '[CLS]' and '[SEP]' in PLMs as a specific entity type. Thus, we have a total of 12 entity types in this work.

### 3.3 Overall Architecture

Figure 2 shows the overall architecture of EARA. Firstly, the retriever module retrieves the nearest instance as a retrieved instance. Secondly, an embedding layer encodes the tokens of the query example and retrieved instance. Next, an entity aligned MHA module encodes the complex semantics of inputs. Finally, a fully-connected layer predicts the similarity score. Figure 3 shows the details of the retriever module and entity aligned MHA module.

### 3.4 Entity-Aligned Multi-Head Attention

Neural machine translation task (Bugliarello and Okazaki, 2020; Slobodkin et al., 2021) shows positive results when introducing external structures information into the attention matrix of transformers. Inspired by this, we introduce an entity alignment matrix constructed from entity information to assist PLMs in learning a more effective attention matrix, thereby facilitating the identification of differences between sentence pairs.

We define an entity alignment matrix $M \in \{0, 1\}^{l \times l}$ as follows:

$$M[i,j] = \begin{cases} 1 & x_i \text{ and } x_j \text{ are the same type,} \\ 0 & \text{otherwise,} \end{cases} \quad (1)$$

where $0 \le i, j < l$, and $l$ is the total length of sentence $S_a$ and $S_b$, i.e., $l = n + m + 3$. $x_i$ and $x_j$ are tokens in sentence $S_a$ and $S_b$. Since $x_0$ is a special token corresponding to '[CLS]' in PLMs, we define $M[i,0]$ or $M[0,j]$ according to the type of the introduced entity information. we set $M[i,0] = 1$ or $M[0,j] = 1$ only in the cases that $x_i$ or $x_j$ is an entity mention. Then, we normalize the entity alignment matrix $M$:

$$V[i,j] = \frac{e^{M[i,j]}}{\sum_{j=0}^{l-1} e^{M[i,j]}}, \quad (2)$$

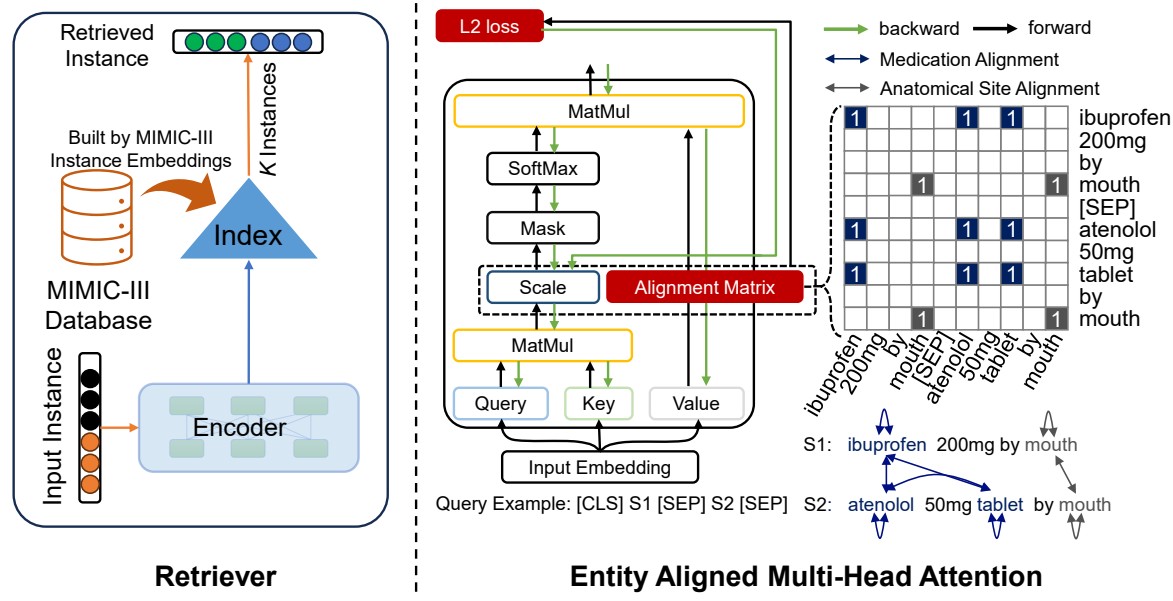

Figure 3: The details of Retriever module and Entity Aligned MHA module.

where $V \in \mathcal{R}^{l \times l}$. As shown in Figure 2, we define an auxiliary L2 loss between multi-head attention scores and alignment matrix to regularize the attention mechanism,

$$L_{align} = \frac{\sum_{h=1}^{H}(V_h - A_h)^2}{H},\qquad(3)$$

where $H$ is the number of aligned attention heads, $V_h$ and $A_h$ are the normalized alignment matrix and the attention weight of the $h$th attention head, respectively. Especially, the entity alignment matrix $V_h$ of each head is equal to $V$. The attention weight $A_h$ is calculated by:

$$A_h = \frac{Q_h^T K_h}{\sqrt{d}},\qquad(4)$$

where $Q_h$ and $K_h$ are query and key in the $h$th attention head, $d$ is the corresponding dimension.

### 3.5 Retriever

The retrieval-augmented module collects the most similar instance to the input from MIMIC-III (Johnson et al., 2016) knowledge store. In this work, we use MIMIC-III NOTEEVENTS.csv as the knowledge database.

#### 3.5.1 MIMIC-III Knowledge Database

To implement the retrieval module, the first step is to construct a knowledge database that can effectively capture the semantics of the input from the MIMIC-III knowledge database $C$. This is achieved by utilizing a frozen BERT encoder $E$

to encode the instances in the MIMIC-III. For each sentence $s_i$ in the MIMIC-III, we generate a key-value pair $(v_i, s_i)$, where $v_i$ represents the embedding of the '[CLS]' token in the final layer of PLMs.

#### 3.5.2 Efficient Searching

Efficient retrieval is crucial considering the potential enormity of the knowledge database. To achieve this, we create a matrix $D \in R^{|C| \times d}$ as the index of MIMIC-III instances, as previously demonstrated in the knowledge database creation process. When a query example is presented during training, we encode it using the encoder $E$ to obtain the query vector $q$. This query vector $q$ is then utilized to search for the $K$ nearest instances in the index $D$ through the maximum inner product search method. To perform this retrieval process, we utilize FAISS (Johnson et al., 2019), an excellent open-source toolkit, to enable fast nearest-neighbour retrieval of MIMIC-III instances from the knowledge store.

### 3.6 Final Similarity Score Prediction

Finally, given the input sentence pair semantic $p$ and nearest retrieved sentence pair $r_0$, they get sentence pair semantic representation $O_p$ and retrieved sentence pair semantic representation $O_{r_0}$. We use a fully-connected layer with ReLU activation to predict the final similarity score $s_p$ and $s_{r_0}$. During training, we minimize the mean square error loss

and alignment loss below:

$$s_p = MLP(O_p), s_r = MLP(O_{r_0}), \quad (5)$$

$$L = L_{sts} + \lambda * L_{align}, \quad (6)$$

$$L_{sts} = MSE(s_p + s_r, s_g), \quad (7)$$

where $s_g$ is the gold score of the input sentence pair, $\lambda$ is a hyper-parameter to weigh the STS prediction loss and alignment loss. During the inference period, we do not use the retrieval instance.

## 4 Experiments

### 4.1 Benchmarks and Metrics

In this section, we give a brief introduction of datasets on which we evaluate all models.

**N2C2STS** (Wang et al., 2020b): The N2C2STS data are expanded on the MedSTS (Wang et al., 2020a) data. All data in the MedSTS dataset are used as the training set for N2C2STS, and 594 additional data are added as the new training set. The training set of N2C2STS has 1,642 sentence pairs in all, and the test set has 412 sentence pairs. The similarity score of N2C2STS ranges from 0 (low similarity) to 5 (high similarity).

**BIOSSES** (Sogancioglu et al., 2017): The BIOSSES comprises 100 sentence pairs, in which each sentence is selected from the Text Analysis Conference (TAC) Biomedical Summarization Track Training Dataset containing articles from the biomedical domain. The data is annotated by five experts and is scored from 0 (no relation) to 4 (equivalent).

**EBMSASS** (Hassanzadeh et al., 2019): EBMSASS consists of 1,000 pairs of clinical evidence. These pairs of clinical evidence are generated from the NICTA-PIBOSO corpus, which is retrieved from 1000 biomedical abstracts. Ten annotators, including eight bioinformatics researchers, one clinical terminology, and one molecular biologist, manually analyze the dataset and assign similarity scores, ranging from 1 (low similarity) to 5 (high similarity).

The BIOSSES and EBMSASS datasets have no specific test set, so we random sample 80% as a training set and 20% are held out as a test set. For in-domain evaluation, following Xiong et al. (Xiong et al., 2020a), we train our supervised model on the training set using a 5-fold cross-validation and evaluate our model on the gold test set. All experimental results are evaluated by the PCC, which measures the linear correlation between two sets of data. PCC is calculated by the ratio between the covariance of two variables and the product of their standard deviations. Therefore, it is basically a normalized measure of covariance such that the result always has a value between -1 and 1.

### 4.2 Baselines and Implementation

We pick 3 representative PLMs as the baseline models and implement our EARA based on these PLMs.

**BERT** (Devlin et al., 2019): BERT is a powerful language model in general domain. It is pretrained on the BooksCorpus (Zhu et al., 2015) and Wikipedia, and reaches state-of-the-art results on the generic natural language processing tasks, like GLUE (Wang et al., 2019).

**BioBERT** (Lee et al., 2020): BioBERT is pretrained with PubMed [2] abstracts and PubMed Central full-text articles based on the pre-trained BERT model weights. It achieves a promising performance on the biomedical NLP task.

**BlueBERT** (Peng et al., 2019): BlueBERT is a pretrained language model based on pre-trained BERT with the addition of PubMed data or electronic medical record MIMIC-III data (Johnson et al., 2016).

We apply the PLMs with 12 layers, hidden size 768 and 12 heads. For the pre-trained PLMs, we tune the learning rate in [1e-5, 2e-5], batch size in [3, 5, 10]. For other parameters, we tune the initial learning rate to be [1e-5, 2e-5], and tune the weighted loss $\lambda$ in [0.1, 0.3, 0.5, 0.7, 1.0]. The maximum sentence length varies from the dataset, 380 for N2C2STS, 200 for BIOSSES and EBMSASS. We run 8 epochs on both datasets. The entire model is implemented with Tensorflow and is trained on 12G GTX 1080 GPU.

### 4.3 Experimental Results

In this section, we evaluate our EARA based on different backbones on the N2C2STS, EBMSASS and BIOSSES and present the results transferred from N2C2STS to EBMSASS and BIOSSES. We compare our results to existing methods and conduct ablation studies.

#### 4.3.1 In-Domain Performance

Table 1 presents a comprehensive summary of the results obtained by our proposed method, EARA,

---

[2]https://pubmed.ncbi.nlm.nih.gov/

|  | N2C2STS | BIOSSES | EBMSASS | Average | Δ% |
|---|---|---|---|---|---|
| BERT | 0.8549 | 0.8570 | 0.9005 | 0.8708 | - |
| EARA (BERT) | 0.8650 | 0.9316 | 0.9244 | 0.9070 | +3.62 |
| BioBERT | 0.8668 | 0.9057 | 0.9181 | 0.8969 | - |
| EARA (BioBERT) | 0.8782 | **0.9446*** | 0.9263 | **0.9163** | +1.94 |
| BlueBERT | 0.8630 | 0.7978 | 0.9226 | 0.8611 | - |
| EARA (BlueBERT) | **0.8872*** | 0.8913 | **0.9313*** | 0.9033 | +4.22 |

Table 1: Pearson Correlation Coefficient (PCC) on different STS datasets with various backbones. **Bolded results** indicate the best performance on each dataset. '*' denotes significant improvement. $\Delta(\%)$ represents the average improvement of the three datasets.

|  | BIOSSES | EBMSASS |
|---|---|---|
| BERT_CLS | 0.5326 | 0.5241 |
| BlueBERT_CLS | 0.7752 | 0.6283 |
| BioBERT_CLS | 0.7759 | 0.6269 |
| BERT | 0.8559 | 0.8266 |
| EARA (BERT) | 0.8604 | 0.8460 |
| BioBERT | 0.8575 | 0.8236 |
| EARA (BioBERT) | **0.8829** | **0.8587** |
| BlueBERT | 0.8403 | 0.8138 |
| EARA (BlueBERT) | 0.8558 | 0.8517 |

Table 2: Out-of-domain performance. **Bolded results** indicate the best results for each model on each dataset.

| Model | N2C2STS |
|---|---|
| (Xiong et al., 2020a) | 0.868 |
| (Ormerod et al., 2021) | 0.870 |
| (Chen et al., 2021) (single) | 0.87 |
| (Mulyar et al., 2021) | 0.867 |
| (Wang et al., 2022b) | 0.875 |
| EARA (BlueBERT) | **0.887** |

Table 3: PCC improvement compared to previous work on the N2C2STS dataset.

across three benchmark datasets: N2C2STS, EBMSASS, and BIOSSES. Notably, our approach consistently outperforms the baseline models, with average improvements ranging from 1.94% to 4.22%. Our best model is significantly better than the baseline model (p-value<0.05). Interestingly, we find that EARA based on BioBERT achieves the highest performance on most datasets. However, our results also demonstrate that EARA, incorporating retrieval augmentation and entity alignment mechanisms, can enable BERT to achieve competitive results with BioBERT and BlueBERT, both of which are pre-trained on domain-specific corpora. This observation suggests that these techniques hold significant potential for enhancing the performance

of BERT in domain-specific tasks. Overall, our findings highlight the effectiveness of our proposed approach and its potential for advancing the state-of-the-art in natural language processing.

### 4.3.2 Out-of-Domain Performance

To demonstrate the generalizability of our proposed model, we trained it on the N2C2STS dataset and evaluated its performance on the entire BIOSSES and EBMSASS datasets. In our evaluation, we compare our approach, EARA, with not-finetuned PLMs along with corresponding baseline PLMs trained on N2C2STS datasets. The experimental results, as presented in Table 2, indicate that our best-performing EARA model outperforms the best not-finetuned PLMs by 9.14% and 23.18% on the BIOSSES and EBMSASS datasets, respectively. Our EARA methods demonstrate promising improvements in the out-of-domain BIOSSES and EBMSASS datasets as compared to the baseline PLMs. Notably, EARA based on BioBERT achieves the highest PCC performance of 0.8829 on the BIOSSES dataset and obtains the best PCC performance of 0.83587 on the EBMSASS dataset. Interestingly, our findings reveal that the model boosting observed on the BIOSSES and EBMSASS datasets is inconsistent with the boosting trend observed on the N2C2STS dataset.

### 4.3.3 Comparison with Prior Works

We compare our model with several published state-of-the-art baselines under the same setting on different STS datasets. As illustrated in Table 3, our EARA based on BlueBERT achieved the highest PCC performance of 0.887 on the N2C2STS dataset, surpassing the previous best model by 1.2%. Notably, the best results for BIOSSES and EBMSASS were reported by (Blagec et al., 2019) and (Wang et al., 2022b), respectively, at 0.871 and 0.922. However, our EARA model outperforms

|           | N2C2   | BIOSSES | EBMSASS |
|-----------|--------|---------|---------|
| EARA(bert) | 0.8650 | 0.9316 | 0.9244 |
| w/o RA    | 0.8593 | 0.8750 | 0.9013 |
| w/o EA    | 0.8640 | 0.8737 | 0.9199 |
| w/o All   | 0.8549 | 0.8570 | 0.9005 |
| EARA(bio) | 0.8782 | 0.9446 | 0.9263 |
| w/o RA    | 0.8726 | 0.9397 | 0.9183 |
| w/o EA    | 0.8793 | 0.9262 | 0.9184 |
| w/o All   | 0.8668 | 0.9057 | 0.9181 |
| EARA(blue) | 0.8872 | 0.8913 | 0.9313 |
| w/o RA    | 0.8699 | 0.8469 | 0.9256 |
| w/o EA    | 0.8750 | 0.8162 | 0.9284 |
| w/o All   | 0.8630 | 0.7978 | 0.9226 |

Table 4: Ablation studies. EA=Entity Alignment matrix; RA=Retrieval Augmentation; w/o=without; bio=BioBERT; blue=BlueBERT. N2C2 is the abbreviation for the N2C2STS dataset, and the brackets indicate which baseline model EARA is based on.

them by a significant margin. Taken together, these experimental results demonstrate the effectiveness of our proposed EARA model and its potential for enhancing the performance of state-of-the-art models in semantic textual similarity tasks.

### 4.3.4 Ablation Study

We conduct ablation studies to assess the individual contributions of the entity alignment matrix and retrieval augmentation to the final results. The results, presented in Table 4, show that removing each part and re-evaluating has a significant impact on the model's performance. Specifically, when we remove the retrieval augmentation module, the decreased performance highlights the importance of adding the logits of retrieved instances during the training period. When we remove the entity alignment matrix module, there is a substantial decrease in model performance, indicating that entity alignment information makes it easier for the model to identify the similarity between two sentence segments. Most notably, when we remove all modules, there is a larger decrease in the model's performance, suggesting that the entity alignment matrix module and retrieval augmentation module complement each other well in enhancing the model's overall performance.

## 5 Discussion

In this section, we conduct further analysis to better understand the results of the experiments.

### 5.1 Error Reduction

Figure 5 analyzes the error rate of EARA in predicting sentences with different numbers of entities. As shown in the figure, when the number of entities in a sentence is greater than 5, compared to Blue-BERT, EARA can significantly reduce the error probability by about 20%. Moreover, the overall error probability is also reduced from 25.7% to 18.9%. Unfortunately, when there are few entities (less than 3) in the sentence, the introduced entity alignment information can slightly increase the error probability. This inspires us to customize the learning strategy based on the numbers of entities in the future, determining when to introduce entity alignment information.

### 5.2 Qualitative Analysis

We visualize the attention weights of baseline PLMs and EARA only with the entity alignment matrix module in Figure 4 to demonstrate that our model provides a clearer explanation. We select sentence pairs from the N2C2STS test sets.

As shown in Figure 4 (a), for the '[CLS]' token, the baseline model (B for '[CLS]') focused more on the first sentence and the punctuation. When we add the entity alignment matrix (A for '[CLS]'), the EARA pays more attention to important tokens, such as medication brands 'ibuprofen' and 'atenolol', which is consistent with human judgment. Figure 4 (b) shows an example without entity alignment information during the inference period. The baseline model puts more attention to stop words and punctuation, but our model learns to focus on important nouns like 'consent', 'testing', 'risk', 'benefit', and 'alternatives' as well as verb information like 'read', 'accepted', 'explained', 'agreed', and 'proceed'.

We also pick a specific token to show the token's contributions. Interestingly, as shown in Figure 4 (a) and (b), the baseline model only focused on the sentence containing the selected token. But with our alignment matrix, EARA can attend to not only the tokens in the sentence containing the selected token but the aligned tokens in the other sentence. Especially in Figure 4 (a), the tokens with the same type as the selected token 'atenolol', such as 'ibuprofen' or 'tablets', are paid more attention. And in Figure 4 (b), the token 'consent' attends to the token 'testing' in the first sentence, and tokens 'explained', 'risk', 'benefit', 'alternatives', 'agreed', and 'proceed' in the second sentence.

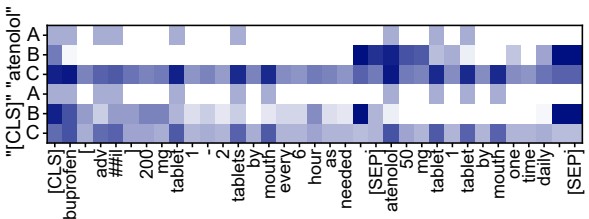
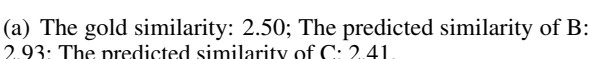
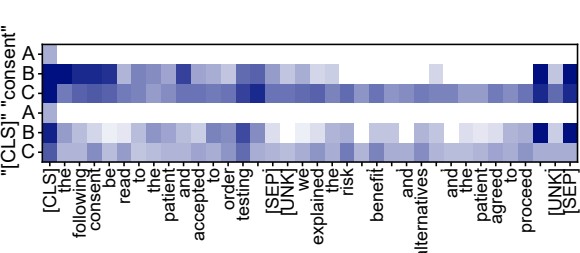

(a) The gold similarity: 2.50; The predicted similarity of B: 2.93; The predicted similarity of C: 2.41.

(b) The gold similarity: 1.00; The predicted similarity of B: 1.91; The predicted similarity of C: 1.54.

Figure 4: Token contribution heatmap. A: entity alignment matrix, B: baseline PLM, C: corresponding EARA. The three continuous rows of A, B, and C is the attention visualization of the token enclosed by quotation on the left.

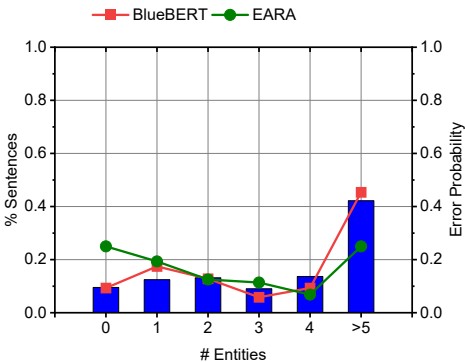

Figure 5: The error probability of EARA(BlueBERT) and BlueBERT on N2C2STS dataset.

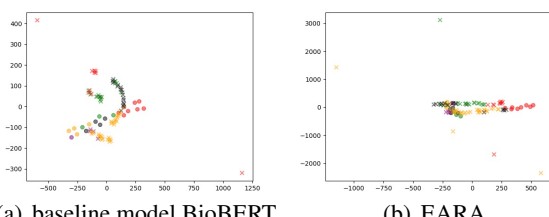

(a) baseline model BioBERT    (b) EARA

Figure 6: The embeddings of the data in a lower-dimensional space using t-SNE of the baseline model and EARA on a very small dataset BIOSSES. 'o' stands for test data and 'x' is the training dataset.

## 5.3 Impact of Entity Number

To investigate the impact of entity information on our proposed EARA model based on BlueBERT, we randomly discard a certain percentage of entity information from the data and evaluate the performance on the N2C2STS dataset using only the entity alignment matrix. Our experimental results reveal that when we randomly mask 0%, 20%, 40%, 60%, 80%, and 100% of entities, the corresponding PCC results are 0.8699, 0.8637, 0.8632, 0.8570, 0.8630, respectively. Notably, when a significant proportion of entity types were removed (i.e., 80% discarded), the entity information may have introduced noise, potentially leading to a decline in the model's performance.

## 5.4 Generalization

To assess the generalization ability of our proposed EARA model, we conduct a visualization analysis of the representations of the training and test sets, as depicted in Figure 6. To generate embeddings, we feed the BIOSSES dataset through the model and extract the '[CLS]' token representation

as the sentence pair representation. We then utilize t-SNE (Van der Maaten and Hinton, 2008) to reduce the dimensionality of the embeddings and visualize them in a 2D space, with similarity scores serving as the labels. Our analysis reveals that the embeddings of the training and test data are closely clustered together in the lower-dimensional space of EARA, indicating that the model is capable of generalizing well to new data.

## 6 Conclusion

In this work, we present an entity-aligned and retrieval augmentation PLM for the biomedical semantic textual similarity tasks. We propose an entity alignment matrix and an auxiliary loss to regularize the attention matrix of PLMs and introduce retrieval augmentation to improve the generalization ability of models further. The comprehensive experiments show that the EARA achieves state-of-the-art performance on the biomedical STS data. Besides, EARA also improves the performance of out-of-domain datasets.

## 7 Limitations

The limitations of our work mainly lie in two aspects. One is that when there are few entities (less than 3) in a sentence, our proposed EARA may hurt the representation of the sentence, leading to a slight increase in prediction error probability, as shown in Figure 5. The other is that the number of retrieved examples is limited by the computation and memory resources. Recent works (Borgeaud et al., 2022a) have demonstrated that introducing more retrievals can improve model's performance.

## 8 Acknowledgement

This study is partially supported by National Key RD Program of China (2021ZD0113402 ), National Natural Science Foundations of China (62276082) and Major Key Project of PCL (PCL2021A06).

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

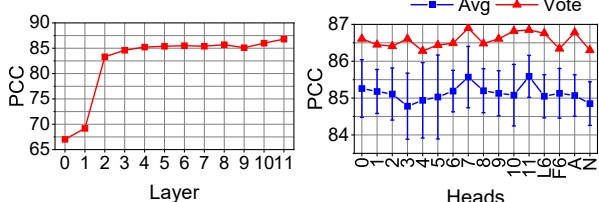

(a) PCC of each transformer layer on N2C2STS.

(b) PCC of supervising different attention heads.

Figure 7: Impact on different transformer layers and different attention heads. L6=Supervising the last 6 heads; F6=Supervising the first 6 heads; A=Supervising all heads; N=Supervising no heads; Avg=Average PCC results, Vote=Voting results.

## A    Choosing Which Layer to Supervise

We evaluate the performance of each layer in Blue-BERT on N2C2STS. We use the '[CLS]' token representation of each layer to predict the similarity score. As shown in Figure 7 (a), we find that the last layer obtains the best PCC performance. Therefore, we choose to supervise self-attention heads of the last layer in PLMs with multi-type fine-grained entity information.

## B    Choosing Which Heads to Supervise

The last layer of PLMs has 12 attention heads, which play different roles (Clark et al., 2019). We investigate which single or multiple heads we should supervise to improve the performance. We evaluate BlueBERT on the N2C2STS and train it with 5-fold cross-validation. We obtain five test results, then we ensemble the results as the voting results to compute the PCC. Besides, we also calculate each test PCC and average the PCC results. When determining which head to regularize, we consider both voting PCC and average PCC.

As shown in Figure 7 (b), we depict the voting PCC and average PCC of five models for different choices of heads as well as the error bar. From the experimental results, we find that supervising the 8th head, the 12th head or all heads can give a better PCC performance. As a result, the results in Table 1 and Table 4 are the results based on the best-performing cases on the validation set among these three scenarios.