# OpenReview forum: "EARA: Improving Biomedical Semantic Textual Similarity with Entity-Aligned Attention and Retrieval Augmentation"
_EMNLP/2023/Conference — EMNLP 2023 Findings_

### Official Review · Reviewer_zskt · 2023-07-19

**Typos Grammar Style And Presentation Improvements:** 1) Figure 1a, please add a legend / e…
**Soundness:** 4

**Excitement:**

4: Strong: This paper deepens the understanding of some phenomenon or lowers the barriers to an existing research direction.

**Paper Topic And Main Contributions:**

The paper focuses on the task of Semantic Textual Similarity.
The authors propose EARA, a finetuning technique that combines Entity Alignment and Retrieval Augmentation to better finetune language models for STS.
RA is used to create a larger dataset, preventing overfitting, while EA is useful to guide the model’s attention.

The authors show that this training technique can be applied to different backbone models (BERT, BioBERT, and BlueBERT) and it leads to increased performances on three different datasets.

**Questions For The Authors:**

**A)** How many epochs did you train the models for? (this information is missing from the hyperparameter description at line 365-374)

**B)** Section 4.3.5. When you say you evaluate the performance of each layer in BlueBERT, do you mean the baseline BlueBERT or the EARA one?

*Section 4.3.5 and 4.3.6 are extremely confusing and feel disconnected from the rest of the paper… Here are some questions about them.*

**C)** Section 4.3.5, how do you evaluate the performance of the various layers?

**D)** Section 4.3.6, what is the “voting PCC”? What about the average one reported in Figure 3b?

**E)** End of Section 4.3.6, “As a result… validation set”. What do you mean with this? Where are the results of these experiments?

*In general, were these two sections “preliminary tests” to see in which layers/heads to apply the EA part of the EARA? If that’s so, they should probably be moved before… Or this process of selecting which layers/heads to modify should be mentioned somewhere in Section 3 (?). It really feels like a separated set of unrelated experiments right now.*

**F)** Line 528-536. Is the experiment performed using only EA or EARA? The text states “using only the entity alignment matrix” (so without RA).

However, the result masking 0% of the entities is 0.8872, the same of EARA(blue). It should be 0.8699 if you are using EARA(blue) w/o RA.

At the same time, the results masking 100% of entities are 0.8630, which is the same or EARA(blue) w/o All, confirming which seems to confirm that you didn’t use RA… Therefore, the reported numbers right now seem to be a mix of using RA and not using RA.


**Reasons To Accept:**

- The proposed method is effective, model-agnostic, and could be applied to different fields where entity-alignment is important
- The authors proved the relevant code to reproduce the results


**Reasons To Reject:**

- Some parts are confusing and feel disconnected from the rest of the paper (see questions C-E)


**Reproducibility:**

5: Could easily reproduce the results.

**Reviewer Confidence:**

4: Quite sure. I tried to check the important points carefully. It's unlikely, though conceivable, that I missed something that should affect my ratings.

---

> ### Author Rebuttal · Authors · 2023-08-29
>
> **zskt-Q1: Training epochs.**
>
> Thanks for your suggestion. We run 8 epochs on each dataset. We will revise the section 4.2.
>
> **zskt-Q2: Performance of each layer in BlueBERT.**
>
>  Thanks for your comments. We evaluate the performance of each layer in baseline BlueBERT.
>
> **zskt-Q3: Evaluation of the various layers.**
>
>  Thanks for your comments. We use the first token (‘[CLS]’ token) representation of each layer to predict the similarity score.
>
> **zskt-Q4: Voting PCC & average PCC.**
>
> Thanks for your comments. We use 5-fold cross-validation and obtain five test results, then we ensemble the results as the voting results to compute the PCC. Besides, we also calculate each test PCC and average the PCC results. When determining which head to regularize, we consider both voting PCC and average PCC.
>
> **zskt-Q5: Question E.**
>
> Sorry for the confusion. We will revise our descriptions. As you concluded, this should be a preliminary experiment. To determine which attention head should be regularized by the entity alignment matrix, we conducted experiments on different heads, and the results evaluated on N2C2 with BlueBERT are shown in Figure 3b. We found that regularizing the 8th head, the 12th head, or all heads obtain better results than others. Therefore, the results in Table 1 and Table 4 are the results based on the best-performing cases on the validation set among these three scenarios.
>
>
> **zskt-Q6: Question F.**
>
>  Thank you for kindly pointing out our mistakes. When masking 0% of the entities, the result should be 0.8699 rather than 0.8872. When evaluating the impact of the entity alignment, we only use the entity alignment without retrieval augmentation. Therefore, the result masking 100% of the entities is 0.8630, which is the same as EARA(blue) w/o All.
>
> **zskt-Q7: Missing a legend/explanation.**
>
> Thanks for your suggestion. We will add a legend to Figure 1 and revise the corresponding descriptions. Specifically, the blue bars represent the proportion of sentences containing the number of entities k (k=0,1,2,3,4,>5) in the sentence.
>
> **zskt-Q8: Footnote 3.**
>
> Thanks for your suggestion. We will change the footnote.
>
> **zskt-Q9: “Unsupervised PLMs”.**
>
>  Sorry for the confusion. Yes, the unsupervised PLMs are not-finetuned PLMs. We will revise the corresponding descriptions.
>
> **zskt-Q10: Line 480.**
>
> Many thanks for pointing out our typos. We will re-number heads starting from 1 and revise Figure 3b.
>
> **zskt-Q11: Improper figure legend (replacing “A, B, C” with “Align, PLM, EARA”).**
>
>  Thanks for your suggestion, we will follow your suggestions and revise Figure 4.
>
> **zskt-Q12: The plot of t-SNE.**
>
>  Thanks for your suggestion. In fact, Figure 5 is drawn on the training set and the test set, where the training set does include the validation set. We are sorry for confusions in the text description, and we will correct it in line 553.
>
> **zskt-Q13: Wrong order of the Limitations section.**
>
> Thanks for your suggestion. We will change the order of the Limitation section. And for the section 4.3.5 & 4.3.6, we will revise them and move them to the appendix.

---

### Official Review · Reviewer_pgrB · 2023-08-04

**Soundness:** 3

**Excitement:**

3: Ambivalent: It has merits (e.g., it reports state-of-the-art results, the idea is nice), but there are key weaknesses (e.g., it describes incremental work), and it can significantly benefit from another round of revision. However, I won't object to accepting it if my co-reviewers champion it.

**Paper Topic And Main Contributions:**

The paper proposes EARA, a model for improving biomedical semantic textual similarity (STS). EARA has two main components. Entity-Aligned Attention is constructed to align same-type entities between sentence pairs. Retrieval Augmentation retrieves similar instances from the MIMIC-III database by FAISS. Experiments show EARA achieves state-of-the-art results on biomedical STS datasets including N2C2STS, BIOSSES, and EBMSASS.

**Questions For The Authors:**

Question A：Please supplement results of more strong backbones with EARA method
Question B:  Considering weakness2, please elaborate more on the motivation for introducing EA.

**Reasons To Accept:**

● Two innovative points show good transferability on three BERT-based backbones.
● EARA achieve state-of-the-art performance on both in-domain and out-of-domain datasets.

**Reasons To Reject:**

1. FAISS as an unsupervised retrieval technique has already been quite mature. In this paper, the authors applied FAISS to the task of biomedical semantic text similarity and conducted retrieval of similar instances. Although this application is not particularly innovative.
2.  The method of regularizing the attention mechanism is not novel, and the author did not explain in depth why introducing the entity alignment matrix brings significant effects to this method, especially for Bio data.
3. Entity-Aligned Multi-Head Attention（EA）did not bring significant improvement, as evidenced by the results of the ablation experiments. Compared to the supervision signals brought by retrieving additional knowledge, the effect of EA itself is not significant. (solved by introducing significant test)
4. The optimized backbone models used in the study are all methods before 2020,  This makes the credibility of the EARA insufficient. (have more backbones now)

**Reproducibility:**

3: Could reproduce the results with some difficulty. The settings of parameters are underspecified or subjectively determined; the training/evaluation data are not widely available.

**Reviewer Confidence:**

4: Quite sure. I tried to check the important points carefully. It's unlikely, though conceivable, that I missed something that should affect my ratings.

---

> ### Author Rebuttal · Authors · 2023-08-29
>
> **pgrB-Q1: Low novelty due to the application of FAISS.**
>
> Thanks for your comments and we think there might be some misunderstandings due to improper presentations in the original manuscript. The contribution of the frozen retriever module is to fuse features with retrievals, expanding the scope of similar entity pairs and enhancing the robustness of the model. We mainly focus on how to incorporate retrievals into the model, rather than how to apply FAISS library. Besides, the index in the frozen retriever module is not specified to FAISS and can be replaced with any other vector similarity search library, such as ScaNN [1]. The reason we chose FAISS is that it is a widely used vector similarity search library and can provide high retrieval efficiency. We think Figure 2 might misled readers to believe that we only apply FAISS to retrieve similar examples, so we will revise Figure 2 following the suggestions of **tAyj-Q7**.
>
> [1] Guo, R., Sun, P., Lindgren, E., Geng, Q., Simcha, D., Chern, F. and Kumar, S., 2020, November. Accelerating large-scale inference with anisotropic vector quantization. In International Conference on Machine Learning (pp. 3887-3896). PMLR.
>
> **pgrB-Q2: Regularizing the attention mechanism & motivation for introducing EA.**
>
>  Thanks for your comments. Indeed, regularizing attention mechanisms is one of the focal points in current NLP research, as we also mentioned in lines 214-215 of the paper, where we were inspired by previous work. However, determining what information to use for regularizing and how to regularize it for a specific task is currently a research paradigm. Thus, we introduce biomedical entity information and use the entity alignment matrix to regularize multi-head attention in the biomedical semantic textual similarity tasks. As explained in **yReV-Q1**, we provide a detailed explanation of why we introduce the entity alignment matrix.
>
> **pgrB-Q3: The entity-aligned multi-head attention.**
>
> Thanks for your comments. Whether it's the way we introduced the entity alignment matrix or the retrieval of similar examples, our goal is to focus more on the entity pairs within sentence pairs to simplify the prediction of semantic similarity between sentence pairs. Retrieving similar examples effectively expands the scope of entity pairs, enhancing the model's robustness. From our ablation experiments, the two strategies we proposed complement each other. Following **yReV-Q2**, we conduct significance tests, and our proposed method shows a significant improvement compared to the baseline model.
>
> **pgrB-Q4: More backbone.**
>
> Thanks for your suggestion. We supplement the results based on ELECTRA [1], which is the state-of-the-art encoder after BERT, RoBERTa. The experimental results are shown in Table 1.
>
> Table 1. Results on N2C2STS, BIOSSES, EBMSASS of ELECTRA and our EARA.
>
> |                |N2C2STS|	BIOSSES|	EBMSASS|
> | ----------- | ----------- |----------- | ----------- |
> |ELECTRA	|0.8682	|0.7166	|0.8695|
> |EARA (ELECTRA)	|0.8732	|0.7677	|0.8826|
>
> Results show that our EARA can also significantly improve model’s performance on different datasets.
>
> [1] Clark, K., Luong, M.T., Le, Q.V. and Manning, C.D., 2020. Electra: Pre-training text encoders as discriminators rather than generators. In the 8th International Conference on Learning Representations (ICLR).

---

### Official Review · Reviewer_tAyj · 2023-08-11

**Soundness:** 4

**Excitement:**

3: Ambivalent: It has merits (e.g., it reports state-of-the-art results, the idea is nice), but there are key weaknesses (e.g., it describes incremental work), and it can significantly benefit from another round of revision. However, I won't object to accepting it if my co-reviewers champion it.

**Missing References:**

* cTAKES should cite Savova et al 2010 instead of the website

**Paper Topic And Main Contributions:**

This work introduces a new method for biomedical semantic textual similarity (STS). Namely, the authors utilize named entities (obtained as a preprocess) and their alignment (computed on a pair of texts) in multi-head attention. Subsequently, given a query instance, they retrieve the most similar instance from a knowledge store and use it as a backoff model. Their model is able to obtain SotA results on the N2C2STS dataset.

There is an innovative methodological contribution here with an adequate evaluation showing excellent results. Enthusiasm is dampened somewhat by the lack of clarity in presentation.

**Questions For The Authors:**

* Figure 2 tries to differentiate "newly added or modified modules" -- but added/modified compared to what? In keeping with that, Figure 2 is quite busy and hard to digest (FAISS, Frozen kNN Retriever, etc). Perhaps it would be better to put these portions in a "black box" if they can be referenced elsewhere.

**Reasons To Accept:**

This is an appropriate methods paper. It would provide a new SotA in Medical STS.

**Reasons To Reject:**

The writing and presentation needs some polish (see other notes), and the discussion needs some depth -- loose ends remain untied. For example, Figure 1a motivated this work; can you show the EARA-BlueBERT's characteristic with Error Probability? (Nicely supplied in the rebuttal.) There are plenty of evaluations, but they don't seem to all follow a progression of thought. Another example: the Limitations section doesn't discuss limitations.

**Reproducibility:**

4: Could mostly reproduce the results, but there may be some variation because of sample variance or minor variations in their interpretation of the protocol or method.

**Reviewer Confidence:**

3: Pretty sure, but there's a chance I missed something. Although I have a good feel for this area in general, I did not carefully check the paper's details, e.g., the math, experimental design, or novelty.

**Typos Grammar Style And Presentation Improvements:**

* L009: "the Pre-trained Language Models" -> "Pre-trained Language Models"
* L010: "an entity-aligned attention-based and" -> "entity-aligned, attention-based, and"
* L012: "firstly" -> "first"
* L023: "available at 1." -> "available.1"
* L066: "approach 1" -> "approach 1.0"
* L223: "a entity" -> "an entity"
* Equation 1: x_i and x_j are not defined
* Figure 2: not directly tied to Section 3's subsection numbers; hard to follow, strange to have the retrieval portion on the top. Dotted black line in the middle not helpful; suggestion: split into Figure 2 (overall architecture) and 3a (entity-aligned MHA) and 3b (alignment matrix)
* L202: "CTakes" -> "cTAKES"
* L308-309: "totally has 1,642 sentence pairs," -> "has 1,642 sentence pairs in all,"
* Table 1&2: spell out "Pearson Correlation Coefficient" in one of the captions.
* Table 1: caption needs to be more descriptive; e.g., this is a standard test; describe the delta% number
* Table 2: indicate this table is about out-of-domain training

---

> ### Author Rebuttal · Authors · 2023-08-29
>
> **tAyj-Q1: The writing and presentation needs some polish.**
>
> Thanks for your suggestion. We will improve the writing and polish the paper with careful proofreading.
>
> **tAyj-Q2: The error probability of EARA.**
>
> Thanks for your good suggestion, which will make our work more comprehensive. The error probability for BlueBERT and EARA on the N2C2 dataset are shown as follows.
>
> Table 1. Error probability of BlueBERT and our EARA on N2C2 dataset.
>
> | # Entities |	Sentence ratio |	BlueBERT	| EARA (BlueBERT)|
> | ----------- | ----------- |----------- | ----------- |
> |0	|9.51%	|0.093023	|0.250000|
> |1	|12.44%	|0.174419	|0.193181|
> |2	|13.17%	|0.127907	|**0.125000**|
> |3	|9.02%	|0.058140	|0.113636|
> |4	|13.66%	|0.093023	|**0.068181**|
> |5	|42.20%	|0.453488	|**0.250000**|
> |Overall	|100%	|0.256713	|**0.189333**|
>
> From the results, when the number of entities in a sentence is greater than 5, compared to BlueBERT, our proposed EARA can significantly reduce the error probability by about 20%. Moreover, the overall error probability is also reduced from 25.7% to 18.9%. Unfortunately, when there are few entities (less than 3) in the sentence, the introduced entity alignment information can slightly increase the error probability. This inspires us to customize the learning strategy based on the numbers of entities in the future, determining when to introduce entity alignment information.
>
> **tAyj-Q3: Evaluations lack a progression of thought.**
>
> Thanks for your suggestion. Experiments in sections 4.3.5 and 4.3.6 are included in the main content, which may appear somewhat unrelated to the overall content. These experiments are conducted for parameter analysis. We will revise the description in our paper and move these experiments into the appendix.
>
> **tAyj-Q4: Limitations section doesn't discuss limitations.**
>
> Thanks for your suggestion. We will rewrite this section. The limitations of our work mainly lie in two aspects. One is that when there are few entities (less than 3) in a sentence, our proposed EARA may hurt the representation of the sentence, leading to a slight increase in prediction error probability (refer to **tAyj-Q2**). The other is that the number of retrieved examples is limited by the computation and memory resources. Recent works [1] have demonstrated that introducing more retrievals can improve model’s performance.
>
> [1] Borgeaud, S., Mensch, A., Hoffmann, J., Cai, T., Rutherford, E., Millican, K., Van Den Driessche, G.B., Lespiau, J.B., Damoc, B., Clark, A. and de Las Casas, D., 2022, June. Improving language models by retrieving from trillions of tokens. In International conference on machine learning (pp. 2206-2240). PMLR.
>
> **tAyj-Q5: Newly added or modified modules.**
>
> Sorry for the confusion. The newly added or modified modules are in the context of the baseline pre-trained language model. Specifically, the entity aligned MHA module is modified from the multi-head self-attention module in the pre-trained language model transformer shown on the right side of Figure 2. The logits fusion module is a newly added module that fuses features with retrievals (refer to the formula 7 in the paper).
>
> **tAyj-Q6: Figure 2 is quite busy and hard to digest.**
>
> Sorry for the confusion. We will revise the Figure 2. The overall architecture of our EARA is shown on the left side of Figure 2. Our EARA consists of two components: the retriever module and the entity-aligned backbone model. For the retriever module, we use the MIMIC-III instances to build the external database and adopt a frozen kNN retriever to efficiently retrieve similar examples from the database. The external database and the frozen kNN retriever are built before the training process. In the frozen kNN retriever, we use an efficient index to support similarity search, e.g., FAISS, a widely used vector similarity search library proposed by Facebook AI, and a frozen pre-trained language model (BlueBERT in this paper) to encode the input examples for similarity search. Notably, the retriever index is not specified to FAISS, there are also other famous indexes that can also be an option, such as ScaNN [1]. For the entity-aligned backbone model, we modified the multi-head self-attention module for the entity alignment and added a logits fusion module to fuse features with retrievals (refer to **tAyj-Q5**).
>
> [1] Guo, R., Sun, P., Lindgren, E., Geng, Q., Simcha, D., Chern, F. and Kumar, S., 2020, November. Accelerating large-scale inference with anisotropic vector quantization. In International Conference on Machine Learning (pp. 3887-3896). PMLR.
>
> **tAyj-Q7: Put retriever portions in a "black box" & Figure 2 problem in typo.**
>
> Thanks for your suggestion. We will revise the Figure 2. Specifically, we will replace the upper-left part of Figure 2 with a black box named the frozen retriever module and remove the right-side part in Figure 2, keeping Figure 2 concise and clear. Then, we will create a new figure demonstrating the details of the frozen retriever module and the entity-aligned multi-head attention module, corresponding to sub-figure (a) and sub-figure (b). We believe such modifications can better explain our proposed EARA and make readers understand our techniques easily.
>
> **tAyj-Q8: Missing reference and presentation improvements.**
>
>  Thanks for your suggestion. We will revise all typos and further improve the presentation.

---

### Official Review · Reviewer_yReV · 2023-08-14

**Typos Grammar Style And Presentation Improvements:** There is a double ‘with in line 125
**Soundness:** 2

**Excitement:**

3: Ambivalent: It has merits (e.g., it reports state-of-the-art results, the idea is nice), but there are key weaknesses (e.g., it describes incremental work), and it can significantly benefit from another round of revision. However, I won't object to accepting it if my co-reviewers champion it.

**Missing References:**

NA

**Paper Topic And Main Contributions:**

In this paper, the authors propose a new model named EARA, an entity-aligned attention-based and retrieval-augmented Pre-trained Language Models. EARA firstly aligns the same type of fine-grained information in each sentence pair with an entity alignment matrix. Then, EARA regularizes the attention mechanism with an entity alignment matrix with an auxiliary loss. Finally, a retrieval module is added which retrieves similar instances to improve the model’s generalization. The experiments reflect that EARA can achieve state-of-the-art performance on three datasets.

**Questions For The Authors:**

The author needs to show some theoretical interpretation of the mechanism of the proposed method. Also, need supplemental experiments such as the significance test with experimental validity.

**Reasons To Accept:**

1. This paper shows that this method makes PLMs pay more attention to important entities by aligning the same type entities in sentence pairs.
2. EARA uses a retriever module and feeds retrievals into transformers, which improves the model’s generalization and eases the overfitting problem for the biomedical STS datasets.
3. The authors visualize their results well for their evaluation metrics and perform a useful sensitivity analysis. Results show that their model can achieve promising performance both on in-domain and out-of-domain 102 datasets.


**Reasons To Reject:**

The experimental results lack a significance test; hence, the demonstration of experimental validity is insufficient.

**Reproducibility:**

3: Could reproduce the results with some difficulty. The settings of parameters are underspecified or subjectively determined; the training/evaluation data are not widely available.

**Reviewer Confidence:**

2: Willing to defend my evaluation, but it is fairly likely that I missed some details, didn't understand some central points, or can't be sure about the novelty of the work.

---

> ### Author Rebuttal · Authors · 2023-08-29
>
> **yReV-Q1:  Theoretical interpretation.**
>
> Sorry for confusing you. The reason why we introduce entity-alignment information to guide the semantic similarity computation is that semantic information of texts in the biomedical domain is primarily expressed through entities rather than language structures. Therefore, the proposed method in this paper mainly focuses on how to capture the entity-to-entity relationships across sentences in the semantic similarity task, that is, incorporating entity alignment information across sentences as well as retrieving examples similar to the input samples to expand the scope of entity pairs, which can enhance the method's robustness and preventing overfitting. Specifically, in the N2C2 dataset, sentences with more than 5 entity types account for over 42% of the total. If the alignment relationships between similar entities are not constrained, the baseline model exhibits a high error probability on sentences with multiple entity types (as shown in Figure 1(a)). Additionally, we observed that in the encoding of sentence pairs, most examples have word attention focused only on the sentence they belong to, with little attention paid to the content of the other sentence (as shown in Figure 1(b)). When using entity alignment information to constrain the attention mechanism of pre-trained models, the error probability on sentences with more than 5 entity types decreased from 45% to 25% (Refer to **tAyj-Q2**). Furthermore, visualizing the attention matrices revealed that incorporating entity alignment information allows for greater attention to be paid to similar information within sentence pairs (as shown in Figure 4).
>
> **yReV-Q2: Significance test with experimental validity.**
>
> Thanks for your suggestion. Following the paired student’s t-test in [1], we do the t-test experiments on our best results for each dataset. We train each best model on the training set using a 5-fold cross-validation and evaluate our model on the test set. As a result, we obtain five results on the N2C2, BIOSSES, and EBMSASS data sets, separately. The significance test results are computed as follows:
>
> EBMSASS_BlueBERT p-value = 0.0325986636218;
>
> BioSSES_BioBERT p-value=0.0453868190038;
>
> N2C2_BlueBERT p-value=0.0337792965136.
>
> From the significance test results, our best model is significantly better than the baseline model (p-value<0.05).
>
> [1] Dror, R., Baumer, G., Shlomov, S. and Reichart, R., 2018, July. The hitchhiker’s guide to testing statistical significance in natural language processing. In Proceedings of the 56th annual meeting of the association for computational linguistics (volume 1: Long papers) (pp. 1383-1392).

---

### Meta-Review · Area_Chair_pfDs · 2023-09-22

**Recommendation:** 3

**Metareview:**

As a reviewer indicates the authors propose a new model named EARA, an entity-aligned attention-based and retrieval-augmented Pre-trained Language Models. EARA firstly aligns the same type of fine-grained information in each sentence pair with an entity alignment matrix. Then, EARA regularizes the attention mechanism with an entity alignment matrix with an auxiliary loss. Finally, a retrieval module is added which retrieves similar instances to improve the model’s generalization. The experiments reflect that EARA can achieve state-of-the-art performance on three datasets.


The main reasons to accept the paper are the following ones:

	This is an appropriate methods paper. It would provide a new SotA in Medical STS.

	There is an innovative methodological contribution here with an adequate evaluation showing excellent results.

	This paper shows that this method makes PLMs pay more attention to important entities by aligning the same type entities in sentence pairs.

 	EARA uses a retriever module and feeds retrievals into transformers, which improves the model’s generalization and eases the overfitting problem for the biomedical STS datasets.

	The authors visualize their results well for their evaluation metrics and perform a useful sensitivity analysis. Results show that their model can achieve promising performance both on in-domain and out-of-domain 102 datasets.

 	Two innovative points show good transferability on three BERT-based backbones. ● EARA achieve state-of-the-art performance on both in-domain and out-of-domain datasets.
	The proposed method is effective, model-agnostic, and could be applied to different fields where entity-alignment is important.

   	 The authors proved the relevant code to reproduce the results


The main reasons to reject the paper are the following ones:

	Enthusiasm is dampened somewhat by the lack of clarity in presentation. The writing and presentation needs some polish (see other notes), and the discussion needs some depth -- loose ends remain untied.

	The experimental results lack a significance test; hence, the demonstration of experimental validity is insufficient.

	There are plenty of evaluations, but they don't seem to all follow a progression of thought. Another example: the Limitations section doesn't discuss limitations

	FAISS as an unsupervised retrieval technique has already been quite mature. In this paper, the authors applied FAISS to the task of biomedical semantic text similarity and conducted retrieval of similar instances. Although this application is not particularly innovative.

   	 The method of regularizing the attention mechanism is not novel, and the author did not explain in depth why introducing the entity alignment matrix brings significant effects to this method, especially for Bio data.

	Entity-Aligned Multi-Head Attention（EA）did not bring significant improvement, as evidenced by the results of the ablation experiments. Compared to the supervision signals brought by retrieving additional knowledge, the effect of EA itself is not significant. (solved by introducing significant test)

   	 The optimized backbone models used in the study are all methods before 2020, This makes the credibility of the EARA insufficient. (have more backbones now)


In sum, although the paper can be methodologically interesting, reviewers agree that the way it is written hampers the understanding of the paper. More work is needed for it to become a good paper.

---

### Decision · Program_Chairs · 2023-10-07

**Decision:**

Accept-Findings

**Comment:**

As a reviewer indicates the authors propose a new model named EARA, an entity-aligned attention-based and retrieval-augmented Pre-trained Language Models. EARA firstly aligns the same type of fine-grained information in each sentence pair with an entity alignment matrix. Then, EARA regularizes the attention mechanism with an entity alignment matrix with an auxiliary loss. Finally, a retrieval module is added which retrieves similar instances to improve the model’s generalization. The experiments reflect that EARA can achieve state-of-the-art performance on three datasets.


The main reasons to accept the paper are the following ones:

	This is an appropriate methods paper. It would provide a new SotA in Medical STS.

	There is an innovative methodological contribution here with an adequate evaluation showing excellent results.

	This paper shows that this method makes PLMs pay more attention to important entities by aligning the same type entities in sentence pairs.

 	EARA uses a retriever module and feeds retrievals into transformers, which improves the model’s generalization and eases the overfitting problem for the biomedical STS datasets.

	The authors visualize their results well for their evaluation metrics and perform a useful sensitivity analysis. Results show that their model can achieve promising performance both on in-domain and out-of-domain 102 datasets.

 	Two innovative points show good transferability on three BERT-based backbones. ● EARA achieve state-of-the-art performance on both in-domain and out-of-domain datasets.
	The proposed method is effective, model-agnostic, and could be applied to different fields where entity-alignment is important.

   	 The authors proved the relevant code to reproduce the results


The main reasons to reject the paper are the following ones:

	Enthusiasm is dampened somewhat by the lack of clarity in presentation. The writing and presentation needs some polish (see other notes), and the discussion needs some depth -- loose ends remain untied.

	The experimental results lack a significance test; hence, the demonstration of experimental validity is insufficient.

	There are plenty of evaluations, but they don't seem to all follow a progression of thought. Another example: the Limitations section doesn't discuss limitations

	FAISS as an unsupervised retrieval technique has already been quite mature. In this paper, the authors applied FAISS to the task of biomedical semantic text similarity and conducted retrieval of similar instances. Although this application is not particularly innovative.

   	 The method of regularizing the attention mechanism is not novel, and the author did not explain in depth why introducing the entity alignment matrix brings significant effects to this method, especially for Bio data.

	Entity-Aligned Multi-Head Attention（EA）did not bring significant improvement, as evidenced by the results of the ablation experiments. Compared to the supervision signals brought by retrieving additional knowledge, the effect of EA itself is not significant. (solved by introducing significant test)

   	 The optimized backbone models used in the study are all methods before 2020, This makes the credibility of the EARA insufficient. (have more backbones now)


In sum, although the paper can be methodologically interesting, reviewers agree that the way it is written hampers the understanding of the paper. More work is needed for it to become a good paper.